# DISTRIBUTED SPECIALIZATION: RARE-TOKEN NEURONS IN LARGE LANGUAGE MODELS

## ABSTRACT

Large language models (LLMs) struggle with representing and generating rare tokens despite their importance in specialized domains. We investigate whether LLMs develop internal specialization mechanisms through discrete modular architectures or distributed parameter-level differentiation. Through systematic analysis of final-layer MLP neurons across multiple model families, we discover that rare-token processing emerges via *distributed specialization*: functionally coordinated but spatially distributed subnetworks that exhibit three distinct organizational principles. First, we identify a reproducible three-regime influence hierarchy comprising highly influential plateau neurons, power-law decay neurons, and minimally contributing neurons which is absent in common-token processing. Second, plateau neurons demonstrate coordinated activation patterns (reduced effective dimensionality) while remaining spatially distributed rather than forming discrete clusters. Third, these specialized mechanisms are universally accessible through standard attention pathways without requiring dedicated routing circuits. Training dynamics reveal that functional specialization emerges gradually through parameter differentiation, with specialized neurons developing increasingly heavy-tailed weight correlation spectra consistent with Heavy-Tailed Self-Regularization signatures. Our findings establish that LLMs process rare-tokens through distributed coordination within shared architectures rather than mixture-of-experts-style modularity. These results provide insights for interpretable model editing, computational efficiency optimization, and understanding emergent functional organization in transformer networks.

## 1 INTRODUCTION

Large language models (LLMs) have achieved remarkable performance across diverse language tasks, yet they consistently struggle with a fundamental challenge: representing and generating rare tokens that appear infrequently in training data (Kandpal et al., 2023; Zhang et al., 2025; Mallen et al., 2023). This limitation is particularly problematic for specialized domains where critical information often resides in the long tail of token frequency distributions (Zipf, 1949; Wyllys, 1981). Recent work has shown that this challenge can lead to model collapse when training on synthetic data with truncated frequency distributions (Dohmatob et al., 2024; Hataya et al., 2023; Bohacek and Farid, 2023).

While external solutions such as retrieval-augmented generation (Lewis et al., 2020), in-context learning (Dong et al., 2022), and non-parametric memory mechanisms (Borgeaud et al., 2022) have been proposed, a critical question remains unanswered: **do LLMs spontaneously develop internal mechanisms specialized for rare token processing during pre-training?** Understanding these internal mechanisms has profound implications for model interpretability, computational efficiency, and our theoretical understanding of how large neural networks organize functional specialization.

This question resonates with insights from cognitive neuroscience on how biological systems balance statistical regularities and exceptions. **Complementary Learning Systems (CLS) theory** (McClelland et al., 1995; Kumaran et al., 2016) explains human learning through dual neural architectures: neocortical systems that extract statistical regularities from frequent experiences, and hippocampal circuits specialized for rapid encoding of novel, infrequent episodes. By analogy, one might expect effective learning systems to require distinct computational strategies for common versus rare events.

Motivated by this perspective, we consider two competing hypotheses for rare-token specialization in LLMs. The **modular hypothesis**, inspired by mixture-of-experts architectures (Shazeer et al., 2017), predicts a discrete functional separation akin to the hippocampal–neocortical divide: spatially clustered neurons with dedicated routing pathways forming distinct "modules" for rare-token processing. In contrast, the **distributed hypothesis** (Rumelhart et al., 1986; Hinton, 1986) proposes that specialization emerges through parameter-level differentiation within shared computational substrates. Under this framework, the same network architecture implements both common and rare token processing through coordinated activation patterns of spatially distributed neurons, similar to how single cortical areas can exhibit different functional modes. With these competing hypotheses, *modular organization* should produce: (i) spatially clustered specialized neurons, (ii) dedicated attention routing pathways, and (iii) discrete activation patterns. *Distributed organization* should instead exhibit: (i) coordinated but spatially scattered neurons, (ii) universal accessibility through standard mechanisms, and (iii) graded, context-dependent specialization.

Mechanistic interpretability studies have shown that transformer neurons can encode interpretable features, from syntactic dependencies (Manning et al., 2020; Finlayson et al., 2021) to semantic concepts (Gurnee et al., 2023; Bricken et al., 2023). Recent findings also suggest frequency-sensitive mechanisms, with neurons modulating prediction confidence based on token rarity (Stolfo et al., 2024). Yet, prior work has primarily focused on individual neuron functions, leaving open the broader organizational principles underlying rare token specialization.

We address this gap by conducting a systematic investigation of how LLMs internally implement rare token specialization. Specifically, we identify neurons with disproportionate influence on rare token prediction and characterize their organizational principles through five complementary analyses:

1. **Hierarchical Influence:** We reveal a reproducible three-regime structure in rare token processing: plateau neurons, power-law decay neurons, and rapid decay neurons where the plateau regime is absent for common tokens.

2. **Activation Coordination:** We show that rare-token neurons exhibit coordinated activation patterns, as quantified by effective dimensionality, despite being spatially scattered.

3. **Spatial Organization:** Using network modularity analysis, we establish that specialized neurons are distributed rather than clustered into discrete modules.

4. **Attention Routing:** We demonstrate that rare tokens access specialized mechanisms through standard attention pathways without dedicated routing.

5. **Functional Specialization:** Using HT-SR spectral analysis, we show that rare-token neurons develop distinct spectral signatures in their learned weight representations.

Taken together, our findings provide the first systematic evidence that **LLMs implement distributed rather than modular specialization for rare token processing**. This result resolves a fundamental open question about functional organization in transformer architectures. Moreover, it highlights how distributed specialization enables flexible, context-sensitive processing while preserving computational efficiency, which diverges from modular separation predicted by CLS-inspired accounts.

## 2 BACKGROUND

### 2.1 TRANSFORMER ARCHITECTURE

In this study, we focus on the Multi-Layer Perceptron (MLP) sublayers. Given a normalized hidden state $x \in \mathbb{R}^{d_{\text{model}}}$ from the residual stream, the MLP transformation is defined as:

$$\text{MLP}(x) = W_{\text{out}}\phi(W_{\text{in}}x + b_{\text{in}}) + b_{\text{out}}, \tag{1}$$

where $W_{\text{in}} \in \mathbb{R}^{d_{\text{mlp}} \times d_{\text{model}}}$ and $W_{\text{out}} \in \mathbb{R}^{d_{\text{model}} \times d_{\text{mlp}}}$ are learned weight matrices, and $b_{\text{in}}, b_{\text{out}}$ are biases. The nonlinearity $\phi$ is typically a GeLU activation. We refer to individual entries in the hidden activation vector $\phi(W_{\text{in}}x + b_{\text{in}})$ as *neurons*, indexed by their layer and position (e.g., `<layer>.<index>`). The activations $n$ represent post-activation values of these neurons. We selected the last layer as it directly projects into the unembedding matrix that produces token probabilities, which creates a computational bottleneck where feature integration must occur (Wei et al., 2022).

## 2.2 Heavy-Tailed Self-Regularization (HT-SR) Theory

Heavy-Tailed Self-Regularization (hereafter *HT-SR*) theory offers a spectral lens on neural network generalization(Martin and Mahoney, 2019; 2021; Lu et al., 2024; Couillet and Liao, 2022). Specifically, consider a neural network with $L$ layers, let $W_i$ denote a weight matrix extracted from the $i$-th layer, where $W_i \in \mathbb{R}^{m \times n}$ and $m \geq n$. We define the correlation matrix associated with $W_i$ as:

$$X_i := W_i^\top W_i \in \mathbb{R}^{n \times n},$$

which is a symmetric, positive semi-definite matrix. The empirical spectral distribution (ESD) of $X_i$ is defined as:

$$\mu_{X_i} := \frac{1}{n} \sum_{j=1}^{n} \delta_{\lambda_j(X_i)},$$

where $\lambda_1(X_i) \leq \cdots \leq \lambda_n(X_i)$ are the eigenvalues of $X_i$, and $\delta$ is the Dirac delta function. The ESD $\mu_{X_i}$ represents a probability distribution over the eigenvalues of the weight correlation matrix, characterizing its spectral geometry.

HT-SR theory proposes that successful neural network training exhibits heavy-tailed spectral behavior in the ESDs of certain weight matrices, due to self-organization toward a critical regime between order and chaos. Such heavy-tailed behavior is captured by various estimators, and particularly informative among which is the power-law (PL) exponent $\alpha_{\text{Hill}}$, who estimates the tail-heaviness of the eigenvalue distribution. Low values of $\alpha_{\text{Hill}}$ (typically $\alpha < 2$) indicate heavy-tailed behavior, often interpreted as signs of functional specialization and self-organized criticality (Yang et al., 2023). A formal definition of $\alpha_{\text{Hill}}$ and the associated estimation procedure is provided in Section 4.5.

## 3 Rare Token neuron analysis framework

### 3.1 Rare Token Neuron Identification

We hypothesize that certain neurons in LLMs specialize for modulating token-level probabilities of *rare tokens*. From a theoretical perspective, such specialization is consistent with sparse coding principles (Olshausen and Field, 1997) and the information bottleneck framework (Tishby et al., 2000), where limited capacity is selectively allocated under uncertainty.

**Ablation methodology** Following Stolfo et al. (2024), we perform targeted ablation experiments in the last MLP layer. For neuron $i$ with activation $n_i$ in the last MLP layer, we define the *mean-ablated activation*:

$$\tilde{x}^{(i)} = x + (\bar{n}_i - n_i)w_{\text{out}}^{(i)}, \tag{2}$$

where $\bar{n}_i$ is the mean activation of neuron $i$ across a reference subset of inputs, and $w_{\text{out}}^{(i)}$ is the corresponding output weight vector. This intervention isolates the contribution of each neuron to token-level predictions. We quantify the influence of a neuron $i$ on rare-token prediction by $\Delta$loss:

$$\Delta\text{loss}(i) = \mathbb{E}_{x \sim \mathcal{D}} \left| \mathcal{L}(\text{LM}(x), x) - \mathcal{L}(\text{LM}(\tilde{x}^{(i)}), x) \right|, \tag{3}$$

where $\mathcal{L}$ is the token-level cross-entropy loss and $\text{LM}(x)$ is the model output. High $\Delta\text{loss}(i)$ indicates neurons with disproportionate influence on rare token prediction. Intuitively, neurons with higher $\Delta$loss have a disproportionate influence on rare-token predictions, making them candidates for rare-token specialization.

**Experimental setup** To evaluate the ablation effects defined above, we sample 25,088 tokens from the C4 dataset (Raffel et al., 2020), focusing on **rare tokens** identified through a two-stage filtering procedure. First, we **remove extremely rare tokens** that fall below the "elbow point" of the frequency distribution, identified using a sliding-window detection method. Tokens in this region are typically noisy and exhibit unstable behavior in our experiments. Second, from the remaining set, we retain tokens with unigram frequencies below the 15th percentile. For comparison, we define

**common tokens** as those above the 15th percentile. This distinction allows us to contrast neuronal influences on low-frequency versus more predictable tokens.

As the definition of "rare-token" is central to this study, we verified the robustness of our findings across multiple percentile thresholds. The qualitative patterns reported here remain qualitatively stable under these variations. It is however worth noticing, that tokens below the frequency elbow point behave idiosyncratically and inconsistently across analyses, motivating their exclusion from our core rare-token set. While the peculiar processing of this naturally defined token group remains an intriguing open question, it lies beyond the scope of the present work.

Given training data availability, ablation experiments are performed on both the Pythia and GPT-2 model families across various parameter scales. Since GPT-2's original training corpus is not publicly available, we approximate token frequencies using the OpenWebTextCorpus (Gokaslan et al., 2019), a widely accepted replication of GPT-2's dataset.

## 3.2 ANALYSIS FRAMEWORK

Our analysis tests the distributed specialization hypothesis through four subsequent complementary investigations, each designed to evaluate specific theoretical predictions about rare token processing mechanisms. We move beyond individual neuron analysis to characterize the organizational principles governing functional specialization in transformer architectures.

**Activation Coordination Patterns**   We begin by examining activation coordination patterns to test whether specialized neurons exhibit coordinated behavior. The distributed specialization hypothesis predicts that functionally coordinated neurons should produce aligned activation patterns, reducing the effective dimension of their collective activation space. In contrast, modular specialization predicts largely independent activation within localized groups.

For each neuron group $\mathcal{G}$, we construct activation vectors across context–token pairs from the C4 dataset. If neurons act independently, these vectors should be nearly linearly independent (Tao and Vu, 2008). On the other hand, coordinated specialization should manifest as alignment in activation patterns. We quantify this through Principal Component Analysis, defining effective dimension as the smallest $d$ such that cumulative variance explained exceeds threshold $\tau = 0.95$:

$$d_{\text{eff}} = \min\left\{ d : \frac{\sum_{i=1}^{d} \lambda_i}{\sum_{j=1}^{N} \lambda_j} \geq \tau \right\} \tag{4}$$

where $\lambda_i$ are eigenvalues of the activation covariance matrix. Distributed specialization predicts lower $d_{\text{eff}}$ for rare-token neurons relative to random controls.

**Spatial Organization Analysis**   To test whether rare-token neurons form localized modules or operate in a distributed manner, we analyze their spatial organization through correlation networks. Modular organization would manifest as spatially clustered groups of strongly interconnected neurons, whereas distributed specialization predicts functional coordination without clear spatial clustering.

Formally, we represent each rare-token neuron $i \in \mathcal{P}$ as a node in an undirected graph $G = (\mathcal{P}, E)$, with edge weights $A_{ij}$ given by the mutual information between activation vectors of neurons $i$ and $j$ across evaluation data:

$$A_{ij} = MI(a_i, a_j) = \sum_{a_i, a_j} p(a_i, a_j) \log \frac{p(a_i, a_j)}{p(a_i)p(a_j)}, \tag{5}$$

We quantify modular structure using the Louvain community detection algorithm, which computes modularity as:

$$Q = \frac{1}{2m} \sum_{ij} \left[ A_{ij} - \frac{k_i k_j}{2m} \right] \delta(c_i, c_j), \tag{6}$$

where $k_i = \sum_j A_{ij}$ is the weighted degree of neuron $i$, $m = \frac{1}{2} \sum_{ij} A_{ij}$ is the total edge weight, and $\delta(c_i, c_j) = 1$ if neurons $i$ and $j$ fall in the same detected community. Larger $Q$ values indicate stronger

modular clustering (Newman, 2006). To evaluate whether rare-token neurons exhibit meaningful modularity, we compare their $Q$ values against size-matched groups of randomly selected neurons. The distributed specialization hypothesis predicts no significant difference between the two.

**Attention Routing Analysis**   We next ask whether rare tokens rely on dedicated attention pathways that selectively route information to rare-token neurons, or whether they instead access these neurons through standard mechanisms. Under modular organization, rare tokens should be funneled through specialized attention heads, while a distributed organization predicts universal accessibility without dedicated routing.

We focus on attention heads in the final two layers ($L-2$ and $L-1$), which directly influence the last MLP layer containing rare-token neurons. For each head $h$, we quantify attention concentration using the Gini coefficient over its attention distribution:

$$\text{Gini}(h) = 1 - 2 \sum_{i=1}^{n} \frac{n+1-i}{n} \cdot \alpha_{h,i}^{\text{sorted}}, \tag{7}$$

where $\alpha_{h,i}^{\text{sorted}}$ represents the attention weights sorted in ascending order. Higher Gini coefficients indicate more concentrated attention patterns and potential specialized routing behavior.

To test for differential routing, we compute Spearman correlations between attention patterns for rare versus common tokens across all heads. The modular hypothesis predicts distinct correlation patterns, whereas the distributed hypothesis predicts alignment between the two. Finally, we assess functional dependence through ablation. For each head $h$, we measure its impact on rare-token neuron activations as:

$$\text{Impact}(h) = \frac{\|\mathbf{a}_{\text{baseline}} - \mathbf{a}_{\text{ablated}(h)}\|_2}{\|\mathbf{a}_{\text{baseline}}\|_2}, \tag{8}$$

where $\mathbf{a}_{\text{baseline}}$ is the activation vector of rare-token neurons under normal operation, and $\mathbf{a}_{\text{ablated}}(h)$ is obtained after zeroing the output of head $h$. To isolate routing effects, we compare individual head ablations against full layer ablations: strong effects from single heads would support specialized routing, while small individual effects combined with strong aggregate effects point to distributed integration.

**Eigen spectrum analysis**   To complement the activation-based analyses, we examine the spectral properties of weight representations associated with rare-token neurons. This analysis is grounded in Heavy-Tailed Self-Regularization (HT-SR) theory 2.2 (Martin and Mahoney, 2021), which predicts that functionally specialized neural units exhibit distinctive heavy-tailed spectral signatures in their weight correlation structures.

For each identified neuron group $\mathcal{G}$ (rare-token neurons and size-matched random control groups), we extract the corresponding slice of the MLP weight matrix $\mathbf{W}_{\mathcal{G}} \in \mathbb{R}^{|\mathcal{G}| \times d}$ and compute the empirical correlation matrix:

$$\boldsymbol{\Xi}_{\mathcal{G}} = \frac{1}{d} \mathbf{W}_{\mathcal{G}} \mathbf{W}_{\mathcal{G}}^{\top}, \tag{9}$$

where $d$ denotes the hidden dimension. The eigenvalue spectrum $\{\lambda_i\}_{i=1}^{|\mathcal{G}|}$ of $\boldsymbol{\Xi}_{\mathcal{G}}$ characterizes the internal dimensionality and correlation structure of the learned representations.

We quantify spectral heaviness using the Hill estimator:

$$\alpha_{\text{Hill}} = \left[ \frac{1}{k} \sum_{i=1}^{k} \log\left(\frac{\lambda_i}{\lambda_k}\right) \right]^{-1}, \tag{10}$$

where $k$ is the number of eigenvalues in the tail region, selected using the Fix-finger method (Yang et al., 2023) to align the threshold $\lambda_k$ with the peak of the eigenvalue density. Lower $\alpha_{\text{Hill}}$ values indicate heavier tails, suggesting more complex learned structures with stronger feature correlations.

HT-SR theory predicts that specialized neurons should exhibit heavier-tailed spectra ($\alpha_{\text{Hill}} < 2$) than randomly selected controls. We test this prediction by comparing $\alpha_{\text{Hill}}$ across groups and model scales in order to provide spectral evidence for functional differentiation that converges with the activation-level findings.

## 4 RESULTS

### 4.1 HIERARCHICAL ORGANIZATION OF RARE TOKEN PROCESSING

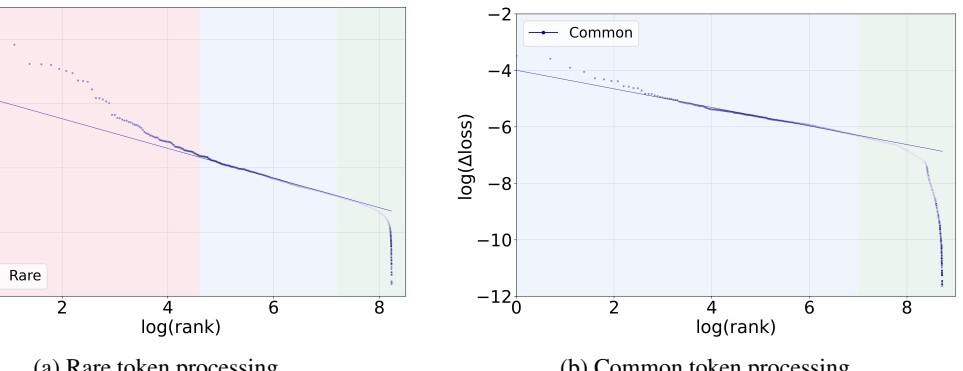

(a) Rare token processing          (b) Common token processing

Figure 1: Neuron influence organization reveals specialization for rare tokens. (a) Rare token processing exhibits a three-regime structure: plateau neurons with exceptional influence (around 1% of total), power-law decay regime, and rapid decay tail. (b) Common token processing shows only smooth power-law decay without plateau structure, demonstrating that specialized mechanisms emerge selectively for rare token processing.

In order to isolate neurons most critical for rare-token prediction, we rank neurons in the final MLP layer by their contribution to loss reduction ($\Delta$Loss). This analysis reveals a consistent organizational structure across model families and scales (Figure 1a 3a).

When ranked by $\Delta$Loss on log–log axes, the distribution follows three regimes as Figure 1a shows:

i.) **Power-law regime** where a group of neurons fall in a power-law regime, manifested as a linear relation between a neuron's $\Delta$Loss and its ranking in log-log coordinates.

$$\log |\Delta\text{Loss}| \approx -\kappa \log(\text{rank}) + \beta, \tag{11}$$

here the power-law exponent $\kappa$ appears as the slope of a linear function;

ii.) **Influential plateau regime** where a small fraction (around 1%) of neurons exhibit consistently higher influence than the power-law regime, forming a plateau in the leftmost region;

iii.) **Rapid decay tail regime** where the majority of uninfluential neurons decay much more rapidly than power-law, indicating negligible contribution to rare token prediction.

This three-regime organization suggests hierarchical resource allocation specifically adapted for rare-token processing. We term the plateau units rare-token neurons, reflecting their exceptional and selective role in rare-token prediction. The rare-token neurons exhibit influence levels that notably exceed what would be expected from uniform or even power-law distributed processing capabilities. Importantly, this regime emerges only for rare tokens: the same analysis on common-token prediction shows no plateau, but instead a smooth power-law decay (Figure 1b). This contrast demonstrates that the three-regime structure reflects targeted specialization, not a generic statistical artifact, and the effect holds across both GPT-2 and Pythia families in different parameter sizes (Figure 3).

| Model | Size | Plateau | Random |
|-------|------|---------|--------|
| Pythia | 1B | 0.79 | 0.88 |
| | 1.4B | 0.74 | 0.89 |
| | 2.8B | 0.72 | 0.89 |
| GPT-2 | 774M | 0.77 | 0.82 |
| | 1.5B | 0.81 | 0.90 |

Table 1: Effective dimensionality across neuron groups. Rare-token neurons consistently exhibit a lower $d_{\text{eff}}$/(number of neurons) ratio than random controls across model architectures and scales, indicating coordinated activation patterns.

| Model | Size | Single | Random | All |
|-------|------|--------|--------|-----|
| Pythia | 1B | 0.34 | 0.37 | -0.50* |
| | 1.4B | 0.28 | 0.32 | -0.45* |
| | 2.8B | 0.30 | 0.34 | -0.47* |
| GPT-2 | 774M | 0.31 | 0.35 | -0.48* |
| | 1.5B | 0.34 | 0.37 | -0.50* |

Table 2: Attention ablation effects across model scales. Only significant effects ($p < 0.05$) are annotated with *. Individual head removals show minimal effects, while ablating all heads produces large changes.

## 4.2 COORDINATED ACTIVATION PATTERNS

We analyze the effective dimensionality of rare-token neuron activation patterns to test for coordinated behavior. These neurons consistently demonstrate lower effective dimensionality than randomly sampled neurons across all examined models (Table 1).

Across Pythia models, plateau neurons consistently show lower dimensionality than random neurons, as seen in Table 1, suggesting that their activations are more linearly dependent and occupy a more constrained subspace. A similar pattern is observed in GPT2-XL, indicating that this low-dimensional structure persists in larger models. This reduction in effective dimensionality points to a shared co-activation pattern among rare-token-selective neurons: rather than acting independently, they tend to activate together in a coordinated manner.

## 4.3 SPATIALLY DISTRIBUTION RATHER THAN CLUSTERING

Community detection analysis reveals that plateau neurons do not exhibit significantly stronger spatial clustering than random baselines (Table 3). Modularity scores show mixed patterns across models: Pythia-70M and Pythia-410M display higher modularity for plateau neurons (0.39 vs. 0.17 and 0.38 vs. 0.15 respectively), while Pythia-2.8B shows a smaller difference (0.34 vs. 0.18, p<0.01). GPT-2 models demonstrate more modest differences, with GPT-2 Large showing 0.31 versus 0.19 and GPT-2 XL showing 0.27 versus 0.18. Only Pythia-2.8B reaches statistical significance for the difference. The inconsistent pattern across models suggests that plateau neurons are spatially distributed rather than forming coherent clusters.

## 4.4 NO EVIDENCE FOR SELECTIVE ATTENTION ROUTING

We next examined whether rare tokens depend on specialized attention pathways that selectively route information into plateau neurons. If such routing existed, we would expect distinct attention signatures for rare tokens and large effects from ablating individual heads.

First, we compared attention distributions between rare and common tokens in the layers preceding the final MLP. The two distributions were nearly indistinguishable: correlations were high ($r = 0.89 \pm 0.07$), and attention concentration did not differ significantly ($G_{\text{rare}} = 0.34 \pm 0.05$ vs. $G_{\text{common}} = 0.32 \pm 0.04$, $p = 0.43$, t-test).

Second, we conducted ablation experiments at the head and layer level (Table 2). Across all model scales, ablating a single attention head produced only small and statistically comparable changes in plateau neuron activation (effect sizes 0.28–0.34), regardless of whether the head was selected for its influence or chosen at random. In contrast, removing all heads in a layer led to large and highly significant activation drops ($-45\%$ to $-50\%$, all $p < 0.001$).

Taken together, these results strongly argue against specialized routing. Rather than relying on a few dedicated heads, plateau neurons integrate signals in a distributed fashion across multiple heads and layers, consistent with the broader distributed coding hypothesis observed in our modularity analyses.

| Model | Size | Group | Modul. | Comm. |
|-------|------|-------|--------|-------|
| Pythia | 1B | Plateau | 0.39 | 10.1 |
| | | Random | 0.17 | 8.1 |
| | 1.4B | Plateau | 0.38 | 8.3 |
| | | Random | 0.15 | 6.6 |
| | 2.8B | Plateau | 0.34* | 7.3 |
| | | Random | 0.18 | 8.3 |
| GPT-2 | 774M | Plateau | 0.31 | 15.0 |
| | | Random | 0.19 | 10.0 |
| | 1.5B | Plateau | 0.27 | 6.6 |
| | | Random | 0.18 | 4.1 |

Table 3: Community detection results across model scales. Modularity scores (Modul.) show inconsistent patterns, indicating distributed rather than clustered organization. Comm. denotes average community size.

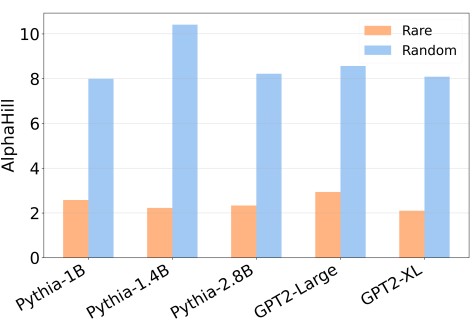

Figure 2: Hill estimator analysis shows consistent spectral differentiation. Plateau neurons exhibit lower $\alpha_{\text{Hill}}$ values than random controls across model families, indicating heavier-tailed weight correlation spectra.

### 4.5 WEIGHT SPECTRAL DIFFERENTIATION

The consistent pattern across model architectures and scales suggests that specialized neurons develop distinct spectral signatures in their learned weight representations.

Analysis of weight correlation matrices using the Hill estimator reveals consistent spectral differences between plateau and random neurons (Figure 2). Plateau neurons exhibit lower $\alpha_{\text{Hill}}$ values compared to random controls across both GPT-2 and Pythia model families, indicating heavier-tailed eigenvalue distributions in their weight correlation matrices, suggesting that spectral differentiation is a fundamental characteristic of rare token processing mechanisms rather than an artifact of specific architectures or parameter scales.

## 5 DISCUSSION

This paper presents a systematic investigation into the neuron mechanisms that transformer language models develop for processing rare tokens, which typically requires a balance between learning and low-frequency generalization. Through converging lines of evidence from neuron influence analysis, activation space geometry, spatial organization, attention routing, and weight spectral analysis, we demonstrate that rare-token competency emerges through parameter-level differentiation within shared computational substrates rather than discrete modular architectures.

Our analysis revealed a three-regime a specialized influential plateau regime, a power-law regime following efficient coding principles, and a rapid decay regime with minimal contribution to rare token processing. This pattern represents hierarchical resource allocation that differs markedly from both uniform processing and strict modular separation. The absence of plateau regime for common tokens highlights that this specialization is not simply a generic feature of network scale but emerges selectively for rare-token processing.

Our activation coordination analysis reveals that plateau neurons operate as functionally integrated units despite spatial distribution across the MLP layer. The consistent reduction in effective dimensionality indicates that these neurons have learned to coordinate their responses in ways that optimize rare-token processing. This coordination occurs without spatial clustering, indicating that functional specialization can emerge through parameter correlation patterns rather than architectural modularity. The universal accessibility of rare-token neurons through standard attention mechanisms reveals that rare-token access specialized processing through the same distributed attention integration used by all tokens rather than requiring specialized routing circuits that would create computational bottlenecks. This design principle enables transformers to maintain flexible, context-sensitive processing while achieving functional specialization.

The heavy-tailed weight correlation spectra observed in rare-token neurons align with Heavy-Tailed Self-Regularization theory predictions about functional specialization. Lower Hill estimator values indicate that specialized neurons develop more complex internal correlation structures, suggesting that meaningful feature learning emerges through spectral differentiation. This provides a bridge between activation-based observations of coordination and weight-based signatures of specialization.

Our findings connect to frameworks in computational neuroscience. On the one hand, our results provide empirical support for distributed representation theory while offering new insights into how artificial neural networks can implement complementary learning principles. The observed three-regime structure echoes Complementary Learning Systems (CLS) theory, where specialized units handle exceptional cases while distributed systems process statistical regularities. However, our results reveal that transformers achieve this dual functionality within a single architecture through parameter differentiation rather than anatomically distinct systems. On the other hans, current findings reflect sparse coding principles in computational neuroscience (Olshausen and Field, 1997), where efficient resource allocation emerges naturally from statistical structure. The hierarchical influence organization suggests that transformers implement a form of adaptive sparsity where computational resources are allocated proportionally to the information content and difficulty of different token types.

Our finding that specialization strength scales with model size indicates that larger models do not simply memorize more patterns, but develop more sophisticated functional differentiation. This provides a mechanistic explanation for improved rare token performance with scale and suggests that continued scaling may yield increasingly refined specialization mechanisms.

For model editing and alignment, our findings suggest that interventions should target coordinated neuron groups rather than individual units. The distributed nature of specialization implies that effective modifications may require understanding and manipulating entire subnetworks rather than localized components.

## 6 FUTURE DIRECTIONS

**Towards deeper MLP layers**   Our analysis focuses exclusively on neurons in the last MLP layer, where feature integration directly impacts token prediction. Rare-token processing, however, may involve interactions across multiple layers and attention mechanisms. Extending the analysis to earlier MLP layers, attention heads, and residual streams would offer a more comprehensive understanding of how LLMs coordinate distributed and specialized processing for low-frequency tokens.

**Applicability to downstream tasks**   We evaluate specialization in the context of next-token prediction on language modeling corpora. The practical impact of these rare-token neurons on downstream applications—such as question-answering, reasoning, or domain-specific generation—remains untested. Future work should examine whether the identified subnetworks meaningfully contribute to task performance and whether interventions on these neurons can improve model efficiency or controllability in applied settings.

The relationship between rare token specialization and other forms of functional organization in transformers requires further investigation. Do similar distributed mechanisms emerge for syntactic processing, semantic relationships, or reasoning tasks? Characterizing the full landscape of functional specialization in large language models represents a frontier for mechanistic interpretability research.

## 7 ETHICS STATEMENT

This work adheres to the ICLR Code of Ethics. In this study, no human subjects or animal experimentation was involved. All datasets used, including C4 dataset, OpenWebTextCorpus and Pile were sourced in compliance with relevant usage guidelines, ensuring no violation of privacy. We have taken care to avoid any biases or discriminatory outcomes in our research process. No personally identifiable information was used, and no experiments were conducted that could raise privacy or security concerns. We are committed to maintaining transparency and integrity throughout the research process.

## 8 REPRODUCIBILITY STATEMENT

We have made every effort to ensure that the results presented in this paper are reproducible. All code and datasets have been made publicly available in an anonymous repository to facilitate replication and verification. The experimental setup, including training steps, model configurations, and hardware details, is described in detail in the paper. We have also provided a full description of ablation experiments, to assist others in reproducing our experiments.

Additionally, C4 dataset, OpenWebTextCorpus and Pile, are publicly available, ensuring consistent and reproducible evaluation results.

We believe these measures will enable other researchers to reproduce our work and further advance the field.

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

# A APPENDIX

## A.1 LLM USAGE

Large Language Models (LLMs) were used to aid in the writing and polishing of the manuscript. Specifically, we used an LLM to assist in refining the language, improving readability, and ensuring clarity in various sections of the paper. The model helped with tasks such as sentence rephrasing, grammar checking, and enhancing the overall flow of the text.

It is important to note that the LLM was not involved in the ideation, research methodology, or experimental design. All research concepts, ideas, and analyses were developed and conducted by the authors. The contributions of the LLM were solely focused on improving the linguistic quality of the paper, with no involvement in the scientific content or data analysis.

The authors take full responsibility for the content of the manuscript, including any text generated or polished by the LLM. We have ensured that the LLM-generated text adheres to ethical guidelines and does not contribute to plagiarism or scientific misconduct.

## A.2 CROSS-MODEL VALIDATION OF NEURON INFLUENCE ORGANIZATION

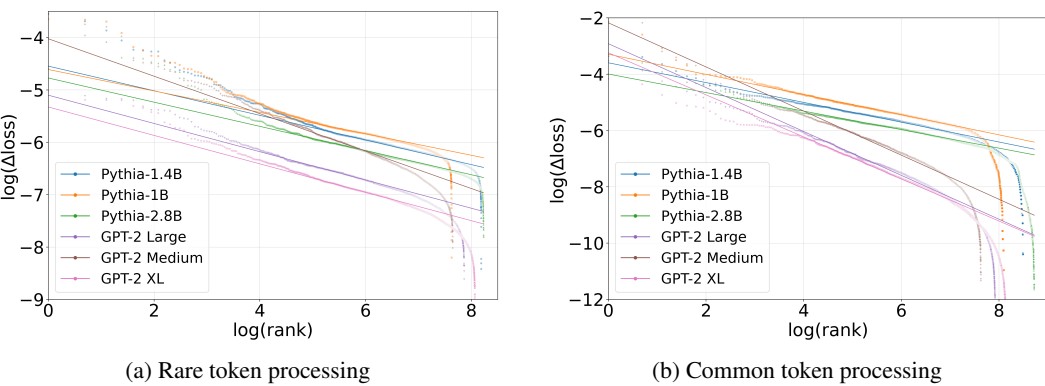

(a) Rare token processing

(b) Common token processing

Figure 3: Neuron influence organization reveals specialization for rare tokens. (a) Rare token processing exhibits a three-regime structure: plateau neurons with exceptional influence , power-law decay regime, and rapid decay tail. (b) Common token processing shows only smooth power-law decay without plateau structure, demonstrating that specialized mechanisms emerge selectively for rare token processing.

To validate the generalizability of our findings, we extend our neuron influence analysis beyond the primary results presented in the main text. Figure 3 demonstrates that the three-regime organizational structure for rare-token processing is not limited to a single model architecture or parameter scale, but represents a consistent computational principle across diverse transformer-based language models. Our extended analysis encompasses multiple model families including GPT-2 (Medium, Large, XL variants) and Pythia (1.4B, 1B, 2.8B configurations), spanning a range of parameter counts. Across all tested configurations, we observe the same fundamental three-regime structure: the influential plateau regime containing a small proportion of neurons, the intermediate power-law regime, and the rapid decay tail regime. The consistency of this pattern across different architectural implementations and parameter scales suggests that the specialized allocation of computational resources for rare-token processing reflects universal optimization pressures in language model training, rather than architecture-specific artifacts. This cross-model validation strengthens our interpretation that rare-token neurons represent a fundamental organizational principle in transformer-based language models.

