# OpenReview forum: "Distributed Specialization: Rare-Token Neurons in Large Language Models"
_ICLR.cc/2026/Conference — ICLR 2026 Conference Withdrawn Submission_

### Official Review · Reviewer_jNsm · 2025-10-30

**Soundness:** 2
**Presentation:** 2
**Contribution:** 1
**Rating:** 2
**Confidence:** 3

**Summary:**

The paper claims rare tokens are handled by distributed specialization rather than modularity, supported by (i) a "plateau" of highly influential last‑layer MLP neurons for rare tokens, (ii) coordinated but spatially scattered activations, (iii) no dedicated attention routing, and (iv) heavier‑tailed weight spectra for those neurons.

**Strengths:**

1. **Relevant question**: The question of distributed representations is of obvious interest within interpretability. Techniques for dealing with distributed representations such as sparse autoencoders form the backbone of the mainstream interpretability stack as practiced at frontier labs like Anthropic.
2. **Clear, testable framing** in investigating the modular vs. distributed hypothesis.
3. **Cross‑model comparisons** (GPT‑2 and Pythia; Fig. 3 in Appendix, p. 13) hint at robustness across scale.

**Weaknesses:**

The empirical case relies almost entirely on last‑layer analyses, has underspecified methodology (rare token selection, neuron clustering, MI estimation, spectral fitting), weak or inconsistent statistics (no uncertainty, minimal significance testing, mixed modularity results), and presentation/framing issues. As written, the core claim is overstated relative to the evidence.

A. Core claim overreaches the scope of evidence

The conclusion "LLMs process rare tokens via _distributed specialization_ and not modularity" is drawn solely from last‑layer MLP analyses (explicitly chosen "as it directly projects into the unembedding matrix," lines 105–106). However, the absence of modular signatures in this part of the model does not rule out modular circuitry in earlier parts of the model (earlier MLPs, attention blocks, residual stream). As such, the headline claim is stronger than the evidence base.

B. The methodology is underspecified

Rare token selection: I’m mildly concerned about the 15% cutoff for rare tokens. The paper says robustness checks across thresholds were performed, but provides no results; please include these in the appendix. Also, tokens below the "elbow point" are excluded because they behave "idiosyncratically," which likely removes the very cases most diagnostic for rare‑token behavior. I could imagine a stronger case being made for this, but it does require additional experimental evidence.

Neuron clustering: It is unclear how the authors determine where to put the boundaries between the regimes. From what I can tell, the distinction between the plateau and power-law regimes seems arbitrary. If I squint when I glance at the common tokens, I believe I can make out some interesting discrete jumps also towards the left-hand side. The y-axis ranges for these tokens are wider, which could mask a slight plateau for common tokens. The evidence showing additional models (Figure 3) provides some weak evidence for a consistent plateau among rare tokens, but this is inconclusive.

All this is to say that the neuron clustering/regime separation methodology requires further specification. Ideally, this process should be automated using e.g. a principled change-point analysis, and additional metrics should be reported arguing for the final regimes as the correct set of regimes.

Mutual information: What probability distributions are being used (5) is missing a definition of the probability distributions? Many other details are missing: are the activations discretized and binned? What is the sample size? MI is very sensitive to these kinds of details. They should be reported and additional ablations should be run.

C.  Statistics and reporting are weak

There are some statistical tests reported in Table 2, but this is otherwise missing from the other parts of the paper. Additionally, there are many different methodologies being invoked, but no hyperparameter ablations reported or included. There are many details of the experimental methodology that should be reported in appendices that are currently missing that not only make it difficult to reproduce but also make it difficult to evaluate whether the techniques are reliable.

D. Presentation and framing issues

The paper is quite difficult to read, largely because of the current split into methodology and results. This split makes sense for most papers, but in this case, it is actually limiting because it requires the reader to remember five different methodologies when going through each results subsection. It would be much more effective to group each methodology together directly with the results.

This presentational challenge reflects a deeper framing/motivation issue. Currently, the paper presents a grab bag of different techniques. The reasons for choosing any particular technique is missing.

There are some other presentation issues:
- The abstract states "Training dynamics reveal that specialization emerges gradually … with increasingly heavy‑tailed spectra" yet **no training‑trajectory plots or checkpoints** are presented in Sec. 4 (only cross‑sectional results). This is a direct mismatch between abstract and results.

- Figure 1: This figure is not clearly self-explanatory. First, the figures should be labeled with what the colors of each stage correspond to. The `rank` can be better explained in the figure or caption. It's not immediately obvious from the figure that the ablated loss is per-neuron. When I first looked at this, I assumed it was per-sample (since this is common for ablations). The stage labels in the caption disagree with the labels and ordering of the regimes in lines 305–316.

- Table 1 (& Sec 4.2): It is unclear from an immediate glance which columns are supposed to be rare vs common tokens. Why are these named according to the names from the regimes in the previous analysis? There is no justification at all for focusing in on regime-specific neurons rather than studying all neurons.

- An open-source repo is mentioned but not provided.

This paper requires significant additional work to strengthen the experimental methodologies and weaken the conclusions drawn from these methodologies. I am unlikely to change my review, barring exceptional changes.

**Questions:**

Questions:
- Line 55: Is there a specific reason to cite mixture-of-experts as the inspiration for the modular hypothesis? From my understanding, the inspiration for looking for modularity goes back much further, both within interpretability and especially within neuroscience.

Suggestions:
- There is currently no mention of superposition/polysemanticity. I would strongly recommend grounding the discussion of modular vs. distributed representations in this existing literature.
- Line 112: (Typo) Missing space between "generalization" and the opening parenthesis.
- Lines 136–138: It is unclear how specialization follows from sparse coding and the IB framework. I would recommend clarifying this.
- Line 443: (Typo)"On the other hans"

---

### Official Review · Reviewer_DVTq · 2025-10-30

**Soundness:** 2
**Presentation:** 2
**Contribution:** 3
**Rating:** 4
**Confidence:** 3

**Summary:**

This work analyses the phenomenon of rare-token processing within the Pythia and GPT-2 large language model (LLM) families, and specifically how such processing is organized within their architecture. It compares two hypotheses: the modular hypothesis, in which dedicated mechanisms and routings exist to process rare tokens, and the distributed hypothesis, in which the same computational substrates process both rare and common tokens. Its primary contribution is the finding that rare-token processing primarily occurs via distributed mechanisms in analyses that span attention heads (Gini coefficients, individual vs. group ablation) and MLP neurons (plateau neuron effective dimensionality and modularity) in the final few layers.

**Strengths:**

1. The authors conduct analyses across multiple model sizes for both Pythia and GPT-2 and show very consistent results for each of their five overarching results (hierarchical influence, activation coordination, spatial organization, attention routing, and functional specialization) across the families and sizes to tackle the distributed vs. modular hypothesis question in language models.

2. The different analyses complement each other well, showing strong breadth. For example, the plateau neurons discovered in Section 4.1 are further analyzed in Section 4.2, 4.3 and 4.5.

**Weaknesses:**

1. While the breadth of analysis is strong, the corresponding depth of each analysis feels shallow at times. First, and most importantly, as the authors acknowledge, this analysis is limited to final layer MLP neurons. As many lines of work show distinct layerwise phenomena, concluding that “LLMs implement distributed rather than modular specialization for rare token processing” (L89-90) may be overgeneralized, representing a significant caveat to the findings.
Additionally, each of the five overarching results is supported by a single diagnostic, without robustness tests or cross-method validation. Further analyses would be beneficial in supporting the claims. For example, extending the activation coordination analysis in Section 4.2 to weight-space similarities or comparing multiple community detection metrics in Section 4.3 could strengthen the claims.

2. Regarding the terminology used, for the spatial organization analysis of MLP neurons, because the index of MLP neurons within a layer has no innate meaning, as the physical location within the brain does, the term “spatial” seems misused. The analysis method itself uses activations as the underlying metric, which is functional rather than spatial. Thus, it seems that this important axis through which to study the distributed vs. modular hypothesis in the original neuroscience context was not truly touched upon within the analyses, unless justified or explained further within the text.

3. In L58-60, the “distributed hypothesis proposes that specialization emerges through parameter-level differentiation within shared computational substrates.” While the text does analyze specialization, further analysis of the notion of sharing beyond the layer level would be appreciated. For example, between rare tokens and common tokens, what is the overlap of the top ranked neurons by their influence? As in, how influential for common tokens is the most influential neuron for rare tokens? Do rare and common tokens share important neurons, not just the entire layer as a whole?

4. Somewhat of a contradiction within the paper is that the fundamental existence of plateau neurons itself seems to suggest some degree of modularity rather than distributive processing. As in, because the plateau neurons seemingly occupy a coordinated subspace based on their activations, that subspace itself represents modular processing. While this is interesting in its own right, the authors should consider presenting and discussing this with more nuance to eliminate any confusion, as concepts extended from neuroscience may not map immediately to transformers
5. The model that the data in Figure 1 is derived from is not specified. Is it from a single model? If so, is there any way to normalize and condense the information in Figure 3 and update Figure 1 with it? Showing data across multiple models in the main text may be more convincing.

Overall, despite the timeliness and breadth of analysis, the analysis is limited to the final layer and the distributed vs modular hypothesis framework requires more nuance and explanation in the context of language models. These points weaken some of the central conclusions of the paper, motivating my score of 4.

**Questions:**

1. The cutoff for the plateau vs. power-law vs. rapid decay regions is not specified. How are the cutoffs determined?
2. Regarding Figure 3, when looking at the GPT-2 family, it seems that, for common tokens, low-rank neurons have lower log delta loss than would be predicted by a power law, almost manifesting as an inversion of the plateau neurons for rare tokens. Do the authors have any insight for this?
3. It is understandable that analyses are conducted on older, densely routed language models due to the availability of their training data token distribution. However, do the authors have any insight on this phenomenon in more modern architectures, such as SwiGLU models? As mixture-of-experts architectures are mentioned, it seems that it would make sense to include such architectures as a control to demonstrate the modular hypothesis.

---

### Official Review · Reviewer_GfPw · 2025-11-01

**Soundness:** 2
**Presentation:** 3
**Contribution:** 2
**Rating:** 4
**Confidence:** 3

**Summary:**

This paper investigates how LLMs process rare tokens. The authors test two competing theories: a “modular hypothesis,” which predicts clustered neurons, and a “distributed hypothesis,” which predicts spatially scattered but functionally coordinated subnetworks. The paper claims that LLMs develop distributed specialization based on an analysis of final-layer MLP neurons and identifies a unique “three-regime” influence hierarchy for rare tokens and finds these neurons are functionally coordinated but spatially distributed.

**Strengths:**

S1: The paper tackles an important problem in mechanistic interpretability; namely, the question of how transformers process rare tokens is of high value to the community. The distinction between the modular and distributed framing is clear and grounded.

S2: The identification of the three-regime structure for rare tokens. The influential plateau provides a clear method for identifying the subnetwork of rare-token neurons, and demonstrating that this plateau vanishes for common-token processing is convincing evidence of targeted specialization.

S3: Showing a high correlation between rare and common token attention patterns and that single-head ablations have little effect whereas full-layer ablations are extremely damaging provides solid evidence towards the distributed theory.

**Weaknesses:**

W1: In Table 3, the caption states “Modularity scores (Modul.) show inconsistent patterns, indicating distributed rather than clustered organization.” This table clearly shows higher modularity scores for Plateau compared to Random which contradicts the paper’s main claim.

W2: These experiments only examine the final layer of the networks so it is an overreach that this implies the distributed hypothesis. It is possible these rare-token neurons are clustered MoE-style in earlier layers of the network and simply aggregated via a distributed network in the final layer.

W3: Excluding the actual rarest tokens misses important data. It’s possible the "distributed" mechanism only applies to "moderately-rare" tokens (those below the 15th percentile but above the elbow) and the modular hypothesis applies to the rarest tokens. This paper’s findings are then limited to the “moderately-rare” tokens which reduces its applicability and scope.

**Questions:**

Q1: Could you please clarify the interpretation in Table 3 taking into account W1 from above?

Q2: Could you justify using mean-ablation over zero-ablation or resample-ablation? Given that specialized rare-token neurons are likely sparse, mean-ablation could be an unnatural choice.

Q3: Are the “rare token neurons” from this work the same as the “token frequency neurons” in Stolfo et al. 2024?

---

### Official Review · Reviewer_pApb · 2025-11-01

**Soundness:** 1
**Presentation:** 2
**Contribution:** 1
**Rating:** 2
**Confidence:** 4

**Summary:**

In this paper the authors investigate how LLMs handle rare tokens. They contrast two competing hypotheses: the modular hypothesis (functional separation) and the distributed hypothesis (“specialization emerges through parameter-level differentiation within shared computational substrates”). Through a variety of metrics they accumulate evidence which suggests that the distributed hypothesis is preferred.

**Strengths:**

Strengths:

* A wide range of metrics are applied to the problem
* A range of models up to 1.5B are examined
* The spectral differentiation does show significant differences between rare and random tokens

**Weaknesses:**

Weaknesses:

* I am not convinced by the distinctions being made between rare and common token processing in Fig 1\. The assertions about a “power law regime” seem to rely entirely on visual inspection of a log-log plot, which is not a valid methodology. There is no attempt made to fit a power law to the relevant segment
* The distinction between the two hypotheses in the introduction is hard for me to follow: terms like “spatially clustered”, “dedicated attention routing pathways” and so on all seem quite informal. Some of them are clarified later in the text, but overall the paper seems to proceed from a somewhat vague characterisation of its main hypothesis to a suite of low-level metrics which are asserted to capture the high level hypotheses. I am not convinced that this is the case, which means I find it hard to assess the significance of the empirical results.

**Questions:**

Questions for the authors:

* How exactly were the boundaries between the colored regions in Fig 1 determined?
* I do not find the references to cognitive neuroscience compelling in the current draft. Could you elaborate on why you have chosen to put this framing so prominently in the introduction?

---

### Note · Authors · 2026-01-15

I have read and agree with the venue's withdrawal policy on behalf of myself and my co-authors.